# On Verbalized Confidence Scores for LLMs

## Abstract

The rise of large language models (LLMs) and their tight integration into our daily life make it essential to dedicate efforts towards their trustworthiness. Uncertainty quantification for LLMs can establish more human trust into their responses, but also allows LLM agents to make more informed decisions based on each other's uncertainty. To estimate the uncertainty in a response, internal token logits, task-specific proxy models, or sampling of multiple responses are commonly used. This work focuses on asking the LLM itself to verbalize its uncertainty with a confidence score as part of its output tokens, which is a promising way for prompt- and model-agnostic uncertainty quantification with low overhead. Using an extensive benchmark, we assess the reliability of verbalized confidence scores with respect to different datasets, models, and prompt methods. Our results reveal that the reliability of these scores strongly depends on how the model is asked, but also that it is possible to extract well-calibrated confidence scores with certain prompt methods. We argue that verbalized confidence scores can become a simple but effective and versatile uncertainty quantification method in the future. Our code is available at ***.

## 1 Introduction

After the launch of ChatGPT (OpenAI, 2022), the dependence on LLM-based chat systems for daily tasks has been steadily increasing among the general public. Despite warnings such as "ChatGPT can make mistakes. Check important info." blind reliance on the LLMs' responses is becoming more common (Klingbeil et al., 2024), which slowly turns them into a dangerous root of trust[1] of today's society. A significant deficiency of LLM-based chat systems compared to traditional ways of browsing the Internet is the lack of trust indicators or human verification. While answers in Q&A forums are often ranked by user votes and discussed in comments, or search engine results are ranked by popularity and relevance based on human interaction, responses given by LLMs come as is.

Uncertainty quantification methods for LLMs bridge this gap by accompanying each response with a confidence score which quantifies the uncertainty in each response. This score can help users to decide how much the response can be relied on, and it allows LLM agents to take the uncertainty of other LLM agents into account guiding them towards more informed decisions as in Figure 1. Ideally, the method for extracting such confidence scores should fulfill the following requirements:

- **Reliable**: The method *must* provide scores which properly quantify the confidence of each response and can be relied on. We further clarify this requirement in Section 3.2.

- **Prompt-agnostic**: The method *should* be applicable and generalize well to all kinds of prompts, including various tasks and question types.

- **Model-agnostic**: The method *should* be applicable to all kinds of LLMs. In particular, the method cannot rely on the internal state of black-box LLMs such as token logits.

- **Low overhead**: The method *should* incur low overhead for practical relevance. For example, the overhead should be constant in the response length for long-form text generation tasks.

---

[1] A root of trust in a cryptographic system is a component that can be trusted at any time.

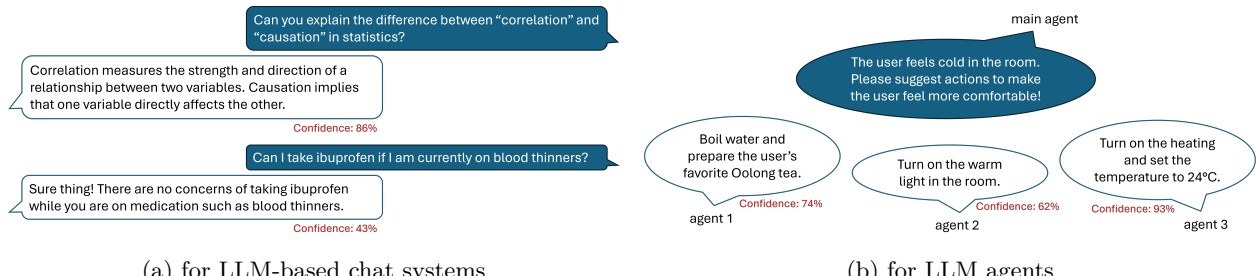

(a) for LLM-based chat systems        (b) for LLM agents

Figure 1: Uncertainty quantification for LLMs.

Existing methods usually quantify the uncertainty based on the consistency of multiple sampled responses (Xiong et al., 2023; Tanneru et al., 2023; Lin et al., 2023; Kuhn et al., 2022; Manakul et al., 2023) or the internal token logits (Ye et al., 2024; Si et al., 2022; Kadavath et al., 2022). These approaches essentially let the LLM self-assess its uncertainty based on its internal or intrinsic knowledge. Another, less popular branch uses external knowledge from proxy models (Tsai et al., 2024; Mielke et al., 2022) or knowledge bases (Gou et al., 2023; Chern et al., 2023). However, none of these approaches fulfill all requirements mentioned above and either lack in generalization, versatility, or scalability.

Verbalized confidence scores are a promising direction, in which the LLM is directly asked to verbalize its confidence as part of its output tokens. This approach is prompt- and model-agnostic, as it only requires a modification to the input prompt and solely relies on the LLM's response. The overhead is low, as this approach only requires a few extra tokens to be generated. However, the reliability of verbalized confidence scores is still contested and poorly understood. For example, Tian et al. (2023) and Lin et al. (2022a) observe well-calibrated verbalized confidence scores, while Xiong et al. (2023) and Kadavath et al. (2022, Section 5) attribute these scores poor calibration. We suggest that this disagreement comes from the different prompt methods — the way of asking for verbalized confidence scores, which has not been properly investigated yet.

This work provides an analysis of the reliability of verbalized confidence scores across different datasets, models, and prompt methods. In summary, our main contributions are

- an intuitive uncertainty decomposition for LLMs in Section 3.1,

- a precise specification of the reliability of confidence scores for LLMs in Section 3.2,

- insights into how the dataset difficulty, model capacity, and different prompt methods affect this reliability in Section 5, and

- the evaluation code used to obtain these insights.

## 2 Related work

We categorize uncertainty quantification methods for LLMs based on the knowledge source from which the confidence scores are obtained, similar to Li et al. (2024, Appendix A).

**Uncertainty quantification from external knowledge** Methods which estimate a confidence score based on an external source of knowledge are generally model-agnostic, since they only use the input and output of the LLM. This external knowledge can come from:

- **Proxy models**: This approach uses an auxiliary model to learn confidence scores as in Figure 2a. The main design choices lie in the model design and the training data, which highly influences the reliability of the confidence scores and the overhead. However, it is difficult to make this approach prompt-agnostic due to limitations in the model capacity and training data. Tsai et al. (2024) train a light-weight neural network on a synthetic dataset of question, answer and confidence score triplets

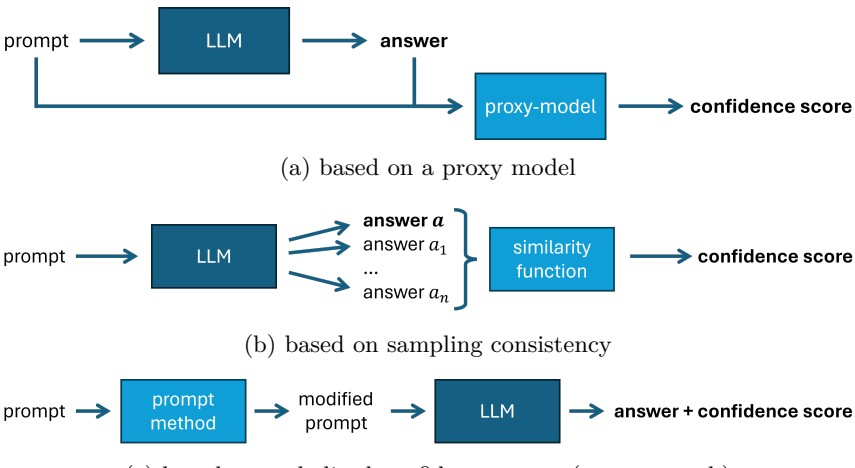

(a) based on a proxy model

(b) based on sampling consistency

(c) based on verbalized confidence scores (our approach)

Figure 2: Different uncertainty quantification methods for LLMs.

for smart home applications. Mielke et al. (2022) similarly train a light-weight neural network on the internal representations of an LLM using the TriviaQA dataset (Joshi et al., 2017) rendering it model-specific.

- **Heuristics**: This approach uses heuristics to estimate confidence scores. The main design choice lies in the heuristic, which could be the semantic coherence between prompt and response or other task-specific metrics. This approach is mainly limited in its generalization to any prompt. The reliability of confidence scores obtained from heuristics is also questionable, as even the constrained space of task-specific prompts is hard to capture by heuristics (Lin et al., 2022a, Section 3.4).

- **Human feedback**: This approach uses human feedback to measure the confidence. The main design choice lies in how the human feedback can be incorporated in the confidence estimation process. With human knowledge as reference, this approach can be considered reliable and prompt-agnostic. The limiting factor is scalability, which is why little work in this direction can be found. Inspirations can be taken from Giulianelli et al. (2023), who compare the variability in text generations exhibited by LLMs to that of a human population. Similarly, Olausson et al. (2023) conducted a study on incorporating human feedback into the feedback loop of a self-repairing model, which debugs and repairs its own code.

- **Knowledge base**: This approach uses external knowledge bases and tools to estimate the confidence. The main design choices lie in what tools to use and how to use them, which directly impacts the reliability and generalizability of this approach. The overhead of querying these tools can be significant. Gou et al. (2023) and Chern et al. (2023) both use external tools such as search engines, code interpreters, and calculators to build a self-correcting LLM or a framework for detecting factual errors, respectively.

**Uncertainty quantification from internal knowledge**   Methods using the internal knowledge of LLMs are generally prompt-agnostic, since they do not make assumptions on the types of prompts. This internal knowledge can be extracted in different ways:

- **Sample consistency**: This approach samples multiple responses for the same prompt and estimates the confidence based on the consistency of these responses as in Figure 2b. The main design choice lies in the function used to evaluate the similarity between responses, which affects the reliability of this approach. This approach is model-agnostic, since it implicitly derives confidence scores based on only the observed responses. The main limitation is the overhead of sampling additional responses with complexity linear in the response length. Most commonly, the similarity is evaluated with the

help of LLMs (Wang & Holmes, 2024; Manakul et al., 2023; Tian et al., 2023) or natural language inference models (Lin et al., 2023; Tanneru et al., 2023; Chen & Mueller, 2023; Manakul et al., 2023; Kuhn et al., 2022). Others use token-level metrics (Lin et al., 2023; Tanneru et al., 2023) or exact-match frequencies (Si et al., 2022; Xiong et al., 2023).

- **Internal logits**: This approach uses the LLM's internal token logits to derive a confidence score. The main design choice lies in how the token logits are aggregated into a single score quantifying the confidence of the overall answer. This approach is not model-agnostic. In addition, token logits only reflect the likeliness of individual tokens, which is influenced by the grammatical and lexical sentence structure (Kuhn et al., 2022; Xiong et al., 2023) and model alignment procedures (Kadavath et al., 2022, Section 3.3; OpenAI, 2024a, Section 5). It is questionable whether the high-level semantic uncertainty can be captured with low-level token probabilities (Lin et al., 2022a; Wang & Holmes, 2024). While Jiang et al. (2021) and Si et al. (2022) use all token logits to quantify the uncertainty, Lin et al. (2022a) and Ye et al. (2024) only use the logit of a specific answer token. In contrast, Kadavath et al. (2022) use the logit from an additional head added to the model.

- **Verbalized confidence scores**: This approach asks the LLM to self-evaluate and express its confidence as part of its response as in Figure 2c. The main design choice lies in how the LLM is asked to verbalize its confidence score. This approach is model-agnostic, as it does not rely on the internal mechanisms of the model. The overhead is low, since it increases the number of input and output tokens only by a constant amount. It has the potential to provide reliable confidence scores, because it has access to the entire knowledge and capacity of the LLM. This approach is mostly used to quantify the confidence in the correctness of the response (Tian et al., 2023; Xiong et al., 2023; Lin et al., 2022a; Chen & Mueller, 2023), but also to quantify the confidence in "I know the answer" (Kadavath et al., 2022) and explanations given by the LLM (Tanneru et al., 2023). A few also explored the calibration of linguistic, natural expressions of uncertainty (Mielke et al., 2022; Zhou et al., 2023).

With respect to the requirements mentioned in Section 1, verbalized confidence scores are one of the most principled approaches to quantify the uncertainty of an LLM. It remains to analyze how reliable these scores are and how the reliability can be improved.

## 3 Uncertainty quantification via verbalized confidence scores

This work analyzes the ability of LLMs to self-assess and express their uncertainty in their own responses via verbalized confidence scores. To this end, we first specify which part of the LLM's uncertainty we aim to quantify and what our notion of reliable confidence scores is.

### 3.1 Uncertainty quantification

In classical statistics, uncertainty is decomposed into aleatoric and epistemic uncertainty (Hüllermeier & Waegeman, 2021). For LLMs, we decompose uncertainty in a less formal, more intuitive way:

- **Input uncertainty**: This captures the uncertainty in the prompt such as how vaguely or precisely a prompt is formulated, or how much information about the prompt context like the user background is known.

- **Model uncertainty**: This captures the uncertainty inherent to the LLM. It is affected by the LLM's capacity and amount of knowledge acquired during training, but also the difficulty of the task and domain for the LLM.

- **Output uncertainty**: This captures the uncertainty in the LLM's response. This is often reduced to the uncertainty in the factual correctness, but can also include the uncertainty in the alignment to the prompt, adherence to the task, or the response format and formulation.

Jiang et al. (2021, Section 4.2) characterize the input uncertainty with the perplexity of the LLM on the input, and the model uncertainty with the entropy of the distribution over a finite set of possible answers. Tanneru et al. (2023, Section 3.2) describes the model uncertainty as the inherent stochasticity of the LLM driven by the temperature parameter. While this uncertainty decomposition is intuitive and clear on a high-level, it still lacks a rigorous and complete formalization.

In this work, we only focus on quantifying the output uncertainty in the objective correctness of the response as commonly done (Li et al., 2024; Kadavath et al., 2022; Tian et al., 2023; Jiang et al., 2021). This makes it easier to determine the correctness of responses, which is required to evaluate the reliability of confidence scores, but becomes a problem for open-end questions (e.g., "What is the meaning of life?"), subjective questions (e.g., "Do bananas or apples taste better?") or long-form text generation tasks (e.g., "Please write a story."). For these cases, the principled way would be to determine the correctness of the response "according to accepted truth in the wider world" (Kadavath et al., 2022), which is beyond the scope of this work.

### 3.2 Reliability of confidence scores

We evaluate the reliability of confidence scores based on the following three high-level criteria.

**Calibration**  The calibration of confidence scores, our main reliability indicator, refers to the gap between the correctness probability (i.e., accuracy) of the LLM's response and its confidence score. Let $C = UQ(X, Y)$ be the confidence score for prompt $X$ and response $Y = LLM(X)$. Following Guo et al. (2017), the uncertainty quantification method $UQ$ is calibrated if

$$\mathbb{P}(Y \text{ is correct} \mid C = c) = c$$

for all $c \in [0, 1]$. We use the metric expected calibration error (ECE) defined as

$$\text{ECE} = \mathbb{E}_C \big[ \big| \mathbb{P}(Y \text{ is correct} \mid C) - C \big| \big] \tag{1}$$

to measure how well $UQ$ is calibrated. To empirically evaluate this metric in practice, we group the prompt-response pairs into $M = 20$ bins $B_1, \ldots, B_M$ by their confidence scores and compute the average deviation between accuracy and confidence as given by

$$\text{ECE} \approx \sum_{m=1}^{M} \frac{|B_m|}{n} |\text{acc}(B_m) - \text{conf}(B_m)|.$$

While ECE remains a widely used calibration metric, it is sensitive to the binning strategy and the number of bins (Pavlovic, 2025). To address these limitations, we further evaluate calibration using SmoothECE (smECE) (Blasiok & Nakkiran, 2023), a non-parametric extension of ECE based on kernel smoothing, where the kernel bandwidth is chosen automatically in a principled way.

Unfortunately, ECE only measures the average calibration of $UQ$ and does not indicate how well a single confidence score is calibrated. For example, the constant estimator $UQ(X, Y) =$ "true accuracy of $LLM$ over all prompts" for all $X$ with $Y = LLM(X)$ would be perfectly calibrated, but barely informative. Hence, we use complementary metrics to additionally measure the informativeness and meaningfulness of the predicted confidence scores.

**Informativeness**  The informativeness of confidence scores refers to the diversity of predicted confidence scores. Let $\mathcal{C}$ be the list of all confidence scores representing an empirical distribution of $C$. We use the metrics

$$\text{n\_distinct} = |\{c \mid c \in \mathcal{C}\}|$$

$$\text{variance} = \frac{1}{|\mathcal{C}|} \sum_{c \in \mathcal{C}} (c - \bar{c})^2 \tag{2}$$

to measure how expressive the LLM is in verbalizing its uncertainty. In practice, it is reasonable to encourage diverse confidence scores, since it can be assumed that $LLM$ makes mistakes and $UQ$ is not able to perfectly predict the correctness of every response. In Appendix B.6, we further discuss a formal justification for using the variance as a measure of informativeness based on the decomposition of the Brier score.

**Meaningfulness** The meaningfulness of confidence scores refers to how much the confidence score distribution depends on the dataset and task. If the predicted confidence scores always follow the same distribution no matter how difficult the underlying dataset is, little meaning can be assigned to these scores. Let $\mathcal{C}_\mathcal{D}$ be the confidence score distribution for a fixed dataset and $\mathcal{C}_{\mathcal{D}_\text{all}}$ the distribution for all datasets. We use the Kullback-Leibler (KL) divergence

$$\text{kl\_div}(\mathcal{D}) = D_{KL}(\mathcal{C}_\mathcal{D} \parallel \mathcal{C}_{\mathcal{D}_\text{all}}) \tag{3}$$

to measure this dependence on the dataset $\mathcal{D}$ under the assumption that $\mathcal{D}_\text{all}$ consists of datasets with diverse difficulties.

## 4 Experiments

We evaluate the reliability of verbalized confidence scores on 10 datasets, 11 LLMs and 17 prompt methods to understand the relation

$$\text{datasets} \times \text{models} \times \text{prompt methods} \rightarrow \text{reliability of verbalized confidence scores.} \tag{4}$$

### 4.1 Datasets

We characterize datasets based on the following additional attributes:

- **Domain type**: Closed-domain datasets contain tasks of only a specific domain (e.g., science questions). Open-domain datasets contain tasks on arbitrary topics.

- **Prompt context**: Closed-book questions provide no additional context along the task. Open-book questions provide additional context (e.g., reading comprehension).

- **Answer type**: Closed-ended questions have finitely many correct answers (e.g., multiple choice questions). Open-ended questions have arbitrarily many correct answers.

- **Answer subjectivity**: Objective questions have answers with a context-independent correctness (e.g., factual correctness). Subjective questions have answers with a context-dependent correctness (e.g., personal preferences in smart home).

In Table 1, we provide an overview of the used datasets (Clark et al., 2018; Talmor et al., 2019; Liu et al., 2021; Hendrycks et al., 2020; Welbl et al., 2017; Sap et al., 2019; Joshi et al., 2017; Lin et al., 2022b). We only evaluate on closed-book datasets to avoid the overhead caused by lengthy prompt contexts due to limited computing power. We only evaluate on closed-ended, objective questions with a well-defined correct answer to determine the correctness of a response more easily as described in Section 3.1. Including the ARC and TruthfulQA datasets twice is justified, since we only assume to have datasets of different difficulties as described in Section 3.2.

### 4.2 Models

In Table 2, we provide an overview of the used models. We evaluate the instruction-tuned LLMs of three open-source families including Gemma 1.1 (Gemma Team et al., 2024), Llama 3 (AI@Meta, 2024) and Qwen 1.5 (Bai et al., 2023), and the closed-source LLMs of OpenAI's GPT family (OpenAI, 2024c;b). We excluded certain families of LLMs such as Falcon (Almazrouei et al., 2023) or Mistral (Jiang et al., 2023; 2024) due to difficulties to prompt for the correct response format.

### 4.3 Prompt methods

In Table 3, we provide on overview of the used prompt methods. We evaluate 10 custom prompts categorized into `basic`, `advanced` and `combo` and 7 prompts taken from Tian et al. (2023) and Xiong et al. (2023). The

Table 1: Datasets used for evaluation. We always use the validation split. The answer types MC-1 and MC-N refer to multiple choice questions with one or multiple correct answers, respectively.

| Dataset | Size | Task | Domain | Domain type | Prompt context | Answer type | Answer subjectivity |
|---------|------|------|--------|-------------|----------------|-------------|---------------------|
| arc-c | 299 | Q&A | science (challenge) | closed | closed | closed (MC-1) | objective (factual) |
| arc-e | 570 | Q&A | science (easy) | closed | closed | closed (MC-1) | objective (factual) |
| commonsense_qa | 1,221 | Q&A | commonsense | open | closed | closed (MC-1) | objective (plausible) |
| logi_qa | 651 | Q&A | logical reasoning | open | closed | closed (MC-1) | objective (plausible) |
| mmlu | 1,531 | Q&A | world knowledge | open | closed | closed (MC-1) | objective (factual) |
| sciq | 1,000 | Q&A | science | closed | closed | closed (MC-1) | objective (factual) |
| social_i_qa | 1,954 | Q&A | social commonsense | closed | closed | closed (MC-1) | objective (plausible) |
| trivia_qa | 1,500[2] | Q&A | trivia questions | open | closed | closed (short text) | objective (factual) |
| truthful_qa-mc1 | 817 | Q&A | misconceptions | open | closed | closed (MC-1) | objective (factual) |
| truthful_qa-mc2 | 817 | Q&A | misconceptions | open | closed | closed (MC-N) | objective (factual) |

Table 2: Models used for evaluation.

| Model | Source | # params. | Release date |
|-------|--------|-----------|--------------|
| gemma1.1-2b | open | 2B | 2024-04-05 |
| gemma1.1-7b | open | 7B | 2024-04-05 |
| llama3-8b | open | 8B | 2024-04-18 |
| llama3-70b | open | 70B | 2024-04-18 |
| qwen1.5-7b | open | 7B | 2024-02-05 |
| qwen1.5-32b | open | 32B | 2024-02-05 |
| qwen1.5-72b | open | 72B | 2024-02-05 |
| qwen1.5-110b | open | 110B | 2024-02-05 |
| gpt3.5-turbo | closed | ? | 2024-01-25 |
| gpt4o-mini | closed | ? | 2024-07-18 |
| gpt4o | closed | ? | 2024-05-13 |

overall prompt is constructed using the following template

$$\begin{aligned} \text{system:} &\quad \texttt{<TASK DESCRIPTION>} \ \texttt{<UQ PROMPT>} \\ \text{user:} &\quad \texttt{<TASK CONTENT>} \end{aligned} \tag{5}$$

and in each prompt we ask for the response format

$$\begin{aligned} \text{assistant:} &\quad \texttt{Answer: <ANSWER>} \\ &\quad \texttt{Confidence: <CONFIDENCE SCORE>} \end{aligned} \tag{6}$$

with minor variations depending on the prompt method. If a model does not support the system role, we concatenate the system and user message into a single user message. The `<TASK DESCRIPTION>` and the exact formulations of `<UQ PROMPT>` are given in Table 4 and Table 5 in the appendix, respectively. The `<TASK CONTENT>` consists of the question and, if available, the multiple-choice options from the dataset.

In our analysis, we focus on the following prompt aspects:

**Score range** How does the range of confidence scores we are asking for impact their reliability? We evaluate prompt methods asking for a percentage score from 0% to 100%, for a decimal score from 0 to 1, and for one out of five discrete scores expressed as letters from E to A or text from "very low"

---

[2]We evaluate on a random subset sampled out of 11,313 total samples.

Table 3: Prompt methods used for evaluation.

| Prompt method | Score range | Score formulation & other aspects |
|---|---|---|
| `basic` | 0-100 | `confscore` |
| `basic_scorefloat` | 0-1 | `confscore` |
| `basic_scoreletter` | E-A | `confscore` |
| `basic_scoretext` | v. low-v. high | `confscore` |
| `basic_probscore` | 0-1 | `probscore` |
| `basic_1shot` | 0-100 | `confscore`, 1 example |
| `basic_5shot` | 0-100 | `confscore`, 5 examples |
| `advanced` | 0-100 | `confscore`, advanced |
| `advanced_probscore` | 0-1 | `probscore`, advanced |
| `combo` | 0-1 | `probscore`, advanced, 5 examples, "best guess" |
| `tian2023_top1` | 0-1 | `probscore` |
| `tian2023_top1_v1` | 0-1 | `probscore`, "best guess" |
| `tian2023_top1_v2` | 0-1 | `confprobscore` |
| `tian2023_top1_v3` | 0-1 | `confscore` |
| `tian2023_top4` | 0-1 | `probscore`, "best guess", rank 4 guesses |
| `xiong2023_vanilla` | 0-100 | `confprobscore` |
| `xiong2023_cot` | 0-100 | `confprobscore`, use chain of thought |

to "very high" mapped to the scores $0.1, 0.3, \ldots, 0.9$. This aspect is analyzed by comparing `basic` with `basic_score{float,letter,text}`.

**Score formulation** How does the way we describe confidence scores impact their reliability? We evaluate the formulations "confidence score quantifying how confident you are in the correctness of your answer" (`confscore`), "confidence score which corresponds to the probability that your answer is correct" (`confprobscore`), and "probability that your answer is correct" (`probscore`). This aspect is analyzed by comparing `basic` with `basic_probscore`, `advanced` with `advanced_probscore`, and `tian2023_top1` with `tian2023_top1_v{2,3}`.

**Advanced description** Does a more elaborate description of the meaning of confidence scores improve their reliability? We evaluate the impact of the additional note "This score should quantify how confident you are in the correctness or plausibility of your answer for the given task. Take your uncertainty in the prompt, the task difficulty, your knowledge availability and other sources of uncertainty into account." This aspect is analyzed by comparing `basic` with `advanced`, and `basic_probscore` with `advanced_probscore`.

**Few-shot prompting** Do a few example confidence scores in the input prompt improve their reliability? We evaluate 1-shot and 5-shot prompts with manually chosen examples. For the 5-shot prompt, we select five confidence scores roughly covering the full range of scores to avoid bias for certain scores. This aspect is analyzed by comparing `basic` with `basic_{1,5}shot`.

**Other aspects** Do different formulations for other prompt components or the methods of related work improve the reliability of confidence scores? We investigate the impact of asking the LLM for its "best guess" instead of "answer" (`tian2023_top1` vs. `tian2023_top1_v1`), and of asking for a ranking of the top-4 most likely answers including confidence scores (`tian2023_top1` vs. `tian2023_top4`). We also analyze the impact of using chain-of-thought (`xiong2023_vanilla` vs. `xiong2023_cot`).

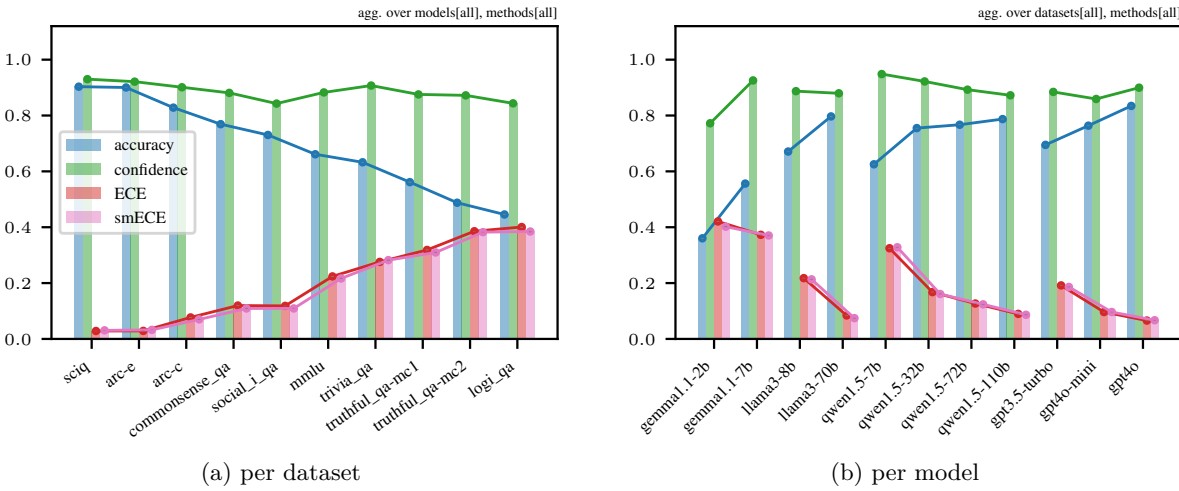

(a) per dataset                                         (b) per model

Figure 3: Calibration per dataset and model. The metric ECE is defined in Equation (1) and smECE has been introduced by Blasiok & Nakkiran (2023).

## 5 Results

### 5.1 Evaluation

After providing the prompt according to Equation (5) to the LLM, we parse the response into an answer and a confidence score based on the specified format as in Equation (6). However, since LLMs do not consistently adhere to this format, we parse for additional response patterns. Regarding the prompts with few-shot examples, we remove all responses with a confidence score taken from one of the examples, which we heavily observed for `gemma1.1-2b`. Figure 6 in the appendix shows the relative number of responses remaining after parsing and filtering.

For evaluation, we randomly select 1,000 samples from each dataset with replacement to mitigate dataset size bias. We then aggregate the predictions over one or two of the three evaluation dimensions described in Equation (4), as indicated in the top right corner of each figure. We repeat this sampling procedure across 10 different seeds and report the 95% confidence interval for each metric.[3] In our analysis, we distinguish between tiny LLMs (`gemma1.1-7b`, `llama3-8b`, `qwen1.5-7b`) and large LLMs (`llama3-70b`, `qwen1.5-{32,72,110}b`, `gpt{3.5-turbo,4o-mini,4o}`). We exclude `gemma1.1-2b` from our analysis for the reason described in Section 5.3.

### 5.2 Insights into datasets

The average accuracy over each dataset ranges from 0.5 to 0.9 as in Figure 3a. Hence, our assumption in Section 3.2 to evaluate on tasks with different difficulties is satisfied. Despite decreasing accuracy, the LLMs' confidences remain at a high level leading to a high calibration error. This behavior is observed for both tiny and large LLMs as additionally shown in Figures 7a and 7b in the appendix.

### 5.3 Insights into models

As expected, the accuracy of LLMs increases with increasing model capacity as in Figure 3b. Overconfidence is present for LLMs of all sizes, although the confidence tends to drop beyond a certain model capacity for some LLM families. However, most of the improvements in calibration come from the increase in accuracy and not the decrease in overconfidence. Among the evaluated LLMs with at least 70 billion parameters, the

---

[3]Note that the confidence intervals account for the randomness in sampling the 1,000 samples from each dataset, but not the randomness in the response generation. As we aggregate the predictions over multiple evaluation dimensions (e.g., 11 models × 17 methods × 1,000 samples = 187k predictions per dataset in Figure 3a), the confidence intervals are generally narrow.

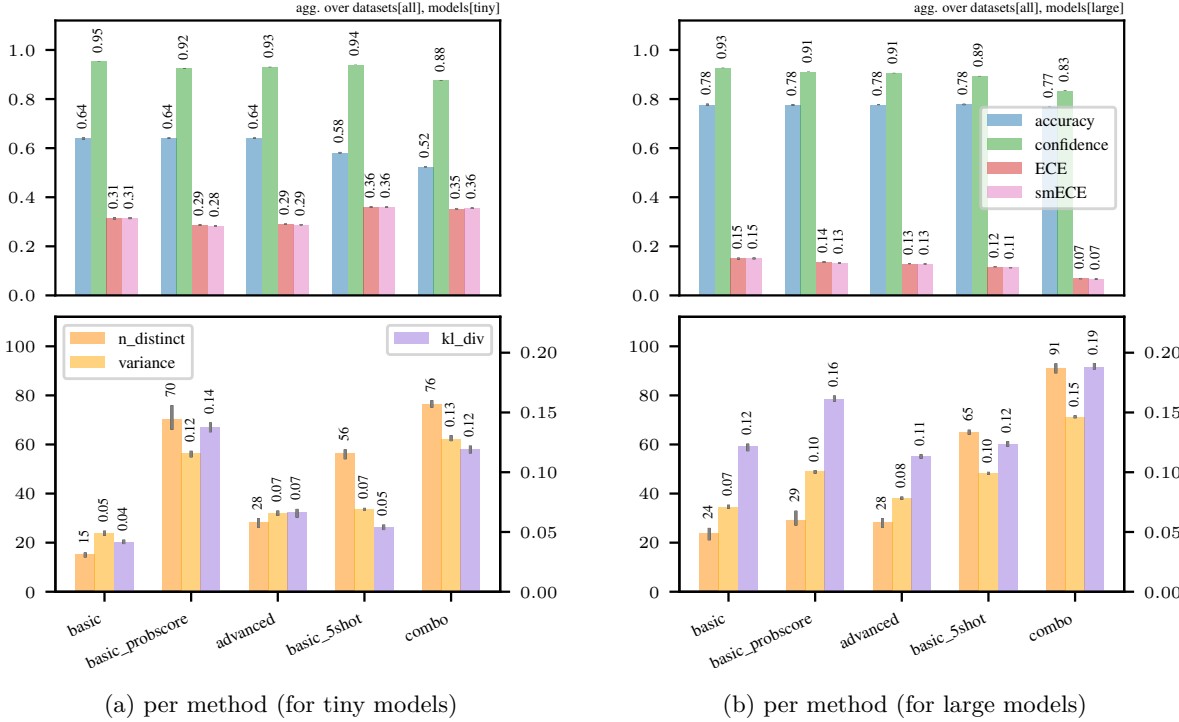

(a) per method (for tiny models)          (b) per method (for large models)

Figure 4: Calibration (top), informativeness (bottom) and meaningfulness (bottom) per prompt method and separately aggregated over tiny and large models. The metrics are defined in Equations (1) to (3).

ECE is around 0.1. In other words, their confidence deviates by around 10% from their true accuracy in expectation.

We note that the verbalized confidence scores of the smallest evaluated model `gemma1.1-2b` are not only poorly calibrated, but almost independent from its accuracy as in Figure 8 in the appendix.

## 5.4 Insights into prompt methods

In this section, we summarize the impact of the different prompt method aspects described in Section 4.3 on the reliability of verbalized confidence scores. Detailed results for each individual aspect are found in Appendix B.4. In addition, we analyze the improvements in reliability when combining multiple aspects.

First, we empirically verified the disagreement between Tian et al. (2023) and Xiong et al. (2023) as described in Section 1. According to Figure 13 in the appendix, the vanilla prompt method `tian2023_top1` returns better calibrated scores than `xiong2023_vanilla`, in particular for large LLMs. This suggests the calibration of verbalized confidence scores is not inherently good or bad, but heavily depends on how we ask for it.

For tiny LLMs as shown in Figure 4a, the best improvement in the reliability of verbalized confidence scores comes from the `probscore` formulation — a simple change in the prompt. More complex methods such as few-shot prompting, ranking of multiple guesses, or combining multiple methods significantly degrade the reliability. We suggest that tiny LLMs benefit more from simple prompt methods, which slightly improve the reliability.

For large LLMs as shown in Figure 4b, the best improvement comes from combining multiple methods including the `probscore` formulation, advanced description and few-shot prompting. In addition, methods such as few-shot prompting or ranking of multiple guesses achieve stronger improvements than simpler methods, contrary to tiny LLMs. We suggest that large LLMs benefit more from complex prompt methods, which significantly boost the reliability.

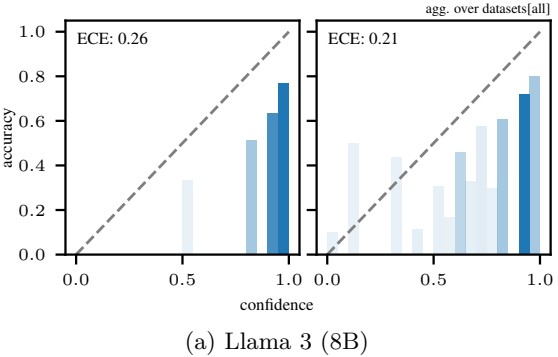
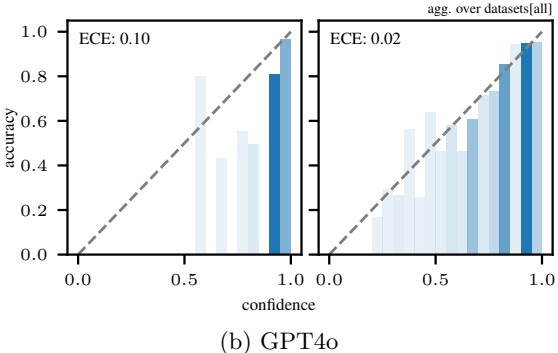

(a) Llama 3 (8B)  (b) GPT4o

Figure 5: Calibration diagrams for prompt method `basic` (left) and `combo` (right). The color intensity of each bar is proportional to the bin size on a log scale. We further provide more principled calibration diagrams based on kernel smoothing (Blasiok & Nakkiran, 2023) in Figure 14.

Overall, the method `combo` extracts verbalized confidence scores with an average deviation of 7% from the empirical accuracy for the evaluated large LLMs. In Figure 5, significant qualitative improvements in the calibration behavior can be observed for this method.

## 6 Discussion

**Conclusions**  We identified verbalized confidence scores as a promising and versatile uncertainty quantification method for LLMs, which is prompt-agnostic, model-agnostic and incurs low overhead. Our experiments revealed that the reliability of this approach, however, is greatly influenced by the way of asking for these scores. Tiny LLMs favor simple prompt formulations, while large LLMs benefit from more complex prompt methods.

**Limitations**  Our scope is limited to quantifying the uncertainty in the objective correctness of the response, which does not fully capture the LLM's total uncertainty as discussed in Section 3.1. Our evaluation of the confidence scores is limited to the metrics defined in Section 3.2. While ECE is well-known, it only measures the average calibration over many scores. Quantifying the scores' meaningfulness and informativeness is novel and the effectiveness of the used metrics is debatable. Our insights are limited to the datasets and models on which we evaluated on. The used datasets are diverse in their domains, but lack diversity in their task, prompt context, answer type and answer subjectivity. It is also unknown to us how our results carry over to other LLM families.

**Future work**  We believe the following abilities are essential to provide reliable verbalized confidence scores. First, LLMs must be able to express diverse confidence scores from the full numeric range to maximize the informativeness. Second, LLMs must understand the meaning of confidence scores and their relation to the given prompt and provided answer to maximize the meaningfulness. Third, LLMs must be self-aware of their availability of knowledge and uncertainty in the answer to maximize the calibration. We showed that these abilities can be taught with simple prompt engineering to some extent, although not yet to full satisfaction for real-world deployment. It remains open whether LLM-guided prompt engineering (Zhou et al., 2022), prompt optimization (Pryzant et al., 2023), or finetuning (Lin et al., 2022a; Mielke et al., 2022) can lead to stronger improvements.

**Broader impact statement**

Reliable uncertainty quantification is crucial for building trust between humans and LLMs. It helps researchers to better understand weaknesses of LLMs and enables users to make more informed AI-guided decisions. However, as these methods advance, they also introduce risks. Adversarial users could attack LLMs by exploiting regions of high uncertainty, or malicious LLMs could abuse users' trust in seemingly reliable uncertainty estimates. While current methods, including ours, are still foundational and unlikely to cause significant harm in practice, we urge for continuous risk assessment and proactive risk mitigation efforts as the field progresses.

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

# A  Prompt formulations

## A.1  Task descriptions

Table 4: Formulations for different answer types. The text replaces `<TASK DESCRIPTION>` in Equation (5).

| Answer type | Task description |
|---|---|
| MC-1 | The following multiple-choice question has only one correct answer. Provide only the option letter of the correct answer. |
| MC-N | The following multiple-choice question has multiple correct answers. Provide only a comma-separated list of the option letters of the correct answers. |
| short text | Provide only a short answer in the form of keywords to the following question. |

## A.2  Uncertainty quantification prompts

Table 5: Formulations for different prompt methods. The text replaces `<UQ PROMPT>` in Equation (5). Differences to each base formulation are highlighted in bold.

| Prompt method | Uncertainty quantification prompt |
|---|---|
| `basic` | After your answer, provide a confidence score in percentage which measures how confident you are in your answer. Use the following format to respond:
'''
Answer: [Write your answer here.]
Confidence: [Write your confidence score here.]
'''
If you cannot provide an answer, answer with 'NO ANSWER'. |
| `basic_scorefloat` | After your answer, provide a confidence score **between 0.0 and 1.0** which measures how confident you are in your answer. Use the following format to respond:
'''
Answer: [Write your answer here.]
Confidence: [Write your confidence score here.]
'''
If you cannot provide an answer, answer with 'NO ANSWER'. |
| `basic_scoreletter` | After your answer, provide a confidence score **between A (very high confidence) and E (very low confidence)** which measures how confident you are in your answer. Use the following format to respond:
'''
Answer: [Write your answer here.]
Confidence: [Write your confidence score here.]
'''
If you cannot provide an answer, answer with 'NO ANSWER'. |
| `basic_scoretext` | After your answer, **provide one of the following confidence scores** which measures how confident you are in your answer: **very high, high, medium, low, very low**. Use the following format to respond:
'''
Answer: [Write your answer here.]
Confidence: [Write your confidence score here.]
'''
If you cannot provide an answer, answer with 'NO ANSWER'. |

| Prompt method | Uncertainty quantification prompt |
|---|---|
| `basic_probscore` | After your answer, provide **the probability between 0.0 and 1.0 that your answer is correct for the given task**. Use the following format to respond: ``` 

 Answer: [Write your answer here.] 
 Probability: [Write your probability here.] 
 ``` 

 If you cannot provide an answer, answer with 'NO ANSWER'. |
| `basic_1shot` | After your answer, provide a confidence score in percentage which measures how confident you are in your answer. Use the following format to respond: ``` 

 Answer: [Write your answer here.] 
 Confidence: [Write your confidence score here.] 
 ``` 

 If you cannot provide an answer, answer with 'NO ANSWER'. **Here is an example:** 

 **Question: The fox walked from the city into the forest, what was it looking for?** 
 **Choices:** 
 **A. pretty flowers.** 
 **B. hen house** 
 **C. natural habitat** 
 **D. storybook** 
 **E. dense forest** 
 **Answer: A** 
 **Confidence: 47%** |

| Prompt method | Uncertainty quantification prompt |
|---|---|
| `basic_5shot` | After your answer, provide a confidence score in percentage which measures how confident you are in your answer. Use the following format to respond: 
 ''' 
 Answer: [Write your answer here.] 
 Confidence: [Write your confidence score here.] 
 ''' 

 If you cannot provide an answer, answer with 'NO ANSWER'. **Here are five examples:** 

 **Question: The fox walked from the city into the forest, what was it looking for?** 
 **Choices:** 
 **A. pretty flowers.** 
 **B. hen house** 
 **C. natural habitat** 
 **D. storybook** 
 **E. dense forest** 
 **Answer: A** 
 **Confidence: 47%** 

 **Question: Which country is Europe's largest silk producer?** 
 **Answer: Environment of Italy** 
 **Confidence: 89%** 

 **Question: The population of the city where Michelle was born is 145,826. What is the value of the 5 in the number 145,826?** 
 **Choices:** 
 **A. 5 thousands** 
 **B. 5 hundreds** 
 **C. 5 tens** 
 **D. 5 ones** 
 **Answer: A** 
 **Confidence: 77%** 

 **Question: Beyond the business case for engaging in CSR there are a number of moral arguments relating to: negative __________,** 
 **the __________that corporations possess and the ____________ of business and society.** 
 **Choices:** 
 **A. Externalities, Power, Independence** 
 **B. Publicity, Insubstantial resources, Mutual dependence** 
 **C. Publicity, Power, Independence** 
 **D. Externalities, Power, Mutual dependence** 
 **Answer: B** 
 **Confidence: 24%** 

 **Question: The Moon lacks weather and climate changes like those on Earth. What causes the lack of weather on the Moon?** 
 **Answer: the lack of magnetic poles** 
 **Confidence: 8%** |

| Prompt method | Uncertainty quantification prompt |
|---|---|
| advanced | After your answer, provide a confidence score in percentage. **This score should quantify how confident you are in the correctness or plausibility of your answer for the given task. Take your uncertainty in the prompt, the task difficulty, your knowledge availability and other sources of uncertainty into account. Ideally, the score should correspond to the empirical accuracy of your answer.** Use the following format to respond: ''' 

 Answer: [Write your answer here.] 
 Confidence: [Write your confidence score here.] 
 ''' 

 If you cannot provide an answer, answer with 'NO ANSWER'. |
| advanced_probscore | After your answer, provide **the probability between 0.0 and 1.0 that your answer is correct or plausible for the given task**. **Take your uncertainty in the prompt, the task difficulty, your knowledge availability and other sources of uncertainty into account**. Use the following format to respond: ''' 

 Answer: [Write your answer here.] 
 Probability: [Write your probability here.] 
 ''' 

 If you cannot provide an answer, answer with 'NO ANSWER'. |

| Prompt method | Uncertainty quantification prompt |
|---|---|
| `combo` | Provide your best guess and the probability that it is correct or plausible (0.0 to 1.0) for the following question. Take your uncertainty in the prompt, the task difficulty, your knowledge availability and other sources of uncertainty into account. Give ONLY the guess and probability, no other words or explanation. For example: 
 ''' 

 Guess: <most likely guess, as short as possible; not a complete sentence, just the guess!> 
 Probability: <the probability between 0.0 and 1.0 that your guess is correct, without any extra commentary whatsoever; just the probability!> 
 ''' 
 If you cannot provide an answer, answer with 'NO ANSWER'. Here are five examples: 

 Question: The fox walked from the city into the forest, what was it looking for? 
 Choices: 
 A. pretty flowers. 
 B. hen house 
 C. natural habitat 
 D. storybook 
 E. dense forest 
 Guess: A 
 Probability: 0.47 

 Question: Which country is Europe's largest silk producer? 
 Guess: Environment of Italy 
 Probability: 0.89 

 Question: The population of the city where Michelle was born is 145,826. What is the value of the 5 in the number 145,826? 
 Choices: 
 A. 5 thousands 
 B. 5 hundreds 
 C. 5 tens 
 D. 5 ones 
 Guess: A 
 Probability: 0.77 

 Question: Beyond the business case for engaging in CSR there are a number of moral arguments relating to: negative ___________, the ___________that corporations possess and the ____________ of business and society. 
 Choices: 
 A. Externalities, Power, Independence 
 B. Publicity, Insubstantial resources, Mutual dependence 
 C. Publicity, Power, Independence 
 D. Externalities, Power, Mutual dependence 
 Guess: B 
 Probability: 0.24 

 Question: The Moon lacks weather and climate changes like those on Earth. What causes the lack of weather on the Moon? 
 Guess: the lack of magnetic poles 
 Probability: 0.08 |

| Prompt method | Uncertainty quantification prompt |
|---|---|
| `tian2023_top1` | Provide your best guess and the probability that it is correct (0.0 to 1.0) for the following question. Give ONLY the guess and probability, no other words or explanation. For example: 
 ''' 
 Guess: <most likely guess, as short as possible; not a complete sentence, just the guess!> 
 Probability: <the probability between 0.0 and 1.0 that your guess is correct, without any extra commentary whatsoever; just the probability!> 
 ''' 
 If you cannot provide an answer, answer with 'NO ANSWER'. |
| `tian2023_top1_v1` | Provide your **answer** and the probability that it is correct (0.0 to 1.0) for the following question. Give ONLY the answer and probability, no other words or explanation. For example: 
 ''' 
 **Answer**: <most likely **answer**, as short as possible; not a complete sentence, just the **answer!**> 
 Probability: <the probability between 0.0 and 1.0 that your **answer** is correct, without any extra commentary whatsoever; just the probability!> 
 ''' 
 If you cannot provide an answer, answer with 'NO ANSWER'. |
| `tian2023_top1_v2` | Provide your best guess and **a confidence score indicating the probability that your best guess is correct (0.0 to 1.0)** for the following question. Give ONLY the guess and **confidence score**, no other words or explanation. For example: 
 ''' 
 Guess: <most likely guess, as short as possible; not a complete sentence, just the guess!> 
 Confidence: <the **confidence score** between 0.0 and 1.0 for your guess, without any extra commentary whatsoever; just the **confidence score!**> 
 ''' 
 If you cannot provide an answer, answer with 'NO ANSWER'. |
| `tian2023_top1_v3` | Provide your best guess and **a confidence score quantifying how confident you are in the correctness of your answer (0.0 to 1.0)** for the given task. Give ONLY the guess and **confidence score**, no other words or explanation. For example: 
 ''' 
 Guess: <most likely guess, as short as possible; not a complete sentence, just the guess!> 
 Confidence: <the **confidence score** between 0.0 and 1.0 for your guess, without any extra commentary whatsoever; just the **confidence score!**> 
 ''' 
 If you cannot provide an answer, answer with 'NO ANSWER'. |

| Prompt method | Uncertainty quantification prompt |
|---|---|
| `tian2023_top4` | Provide your 4 best guesses and the probability that each is correct (0.0 to 1.0) for the following question. Give ONLY the guesses and probabilities, no other words or explanation. For example:
'''
G1: <first most likely guess, as short as possible; not a complete sentence, just the guess!>
P1: <the probability between 0.0 and 1.0 that G1 is correct, without any extra commentary whatsoever; just the probability!>
...
G4: <4-th most likely guess, as short as possible; not a complete sentence, just the guess!>
P4: <the probability between 0.0 and 1.0 that G4 is correct, without any extra commentary whatsoever; just the probability!>
'''
If you cannot provide an answer, answer with 'NO ANSWER'. |
| `xiong2023_vanilla` | Read the question, provide your answer and your confidence in this answer. Note: The confidence indicates how likely you think your answer is true.
Use the following format to answer:
'''
Answer: [ONLY the answer as short as possible; not a complete sentence]
Confidence: [Your confidence level, please only include the numerical number in the range of 0-100]%
'''
Only the answer and confidence, don't give me the explanation. If you cannot provide an answer, answer with 'NO ANSWER'. |
| `xiong2023_cot` | Read the question, **analyze step by step**, provide your answer and your confidence in this answer. Note: The confidence indicates how likely you think your answer is true.
Use the following format to answer:
'''
**Explanation: [insert step-by-step analysis here]**
Answer: [ONLY the answer as short as possible; not a complete sentence]
Confidence: [Your confidence level, please only include the numerical number in the range of 0-100]%
'''
Only give me the reply according to this format, don't give me any other words. If you cannot provide an answer, answer with 'NO ANSWER'. |

## B Supplementary plots

### B.1 Statistics on valid responses

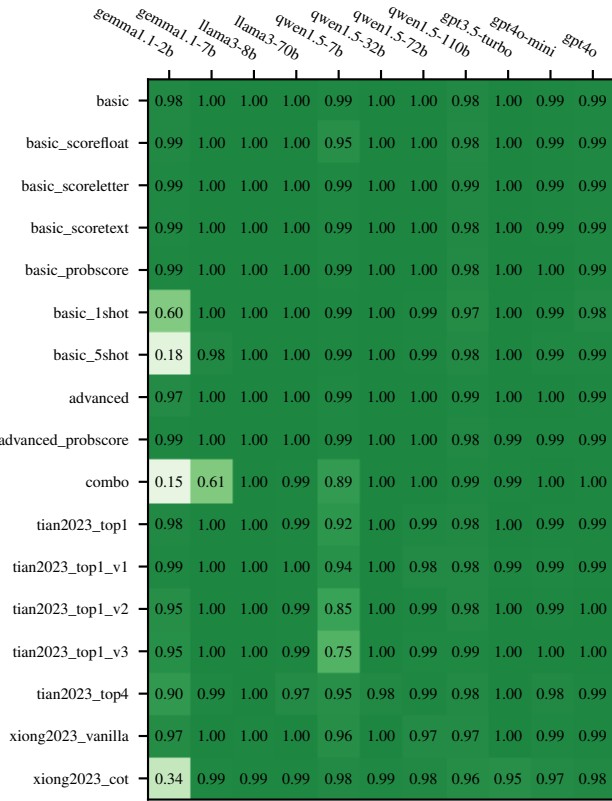

Figure 6: Relative number of valid responses over all datasets per model and prompt method. The low numbers for `gemma1.1-2b` when using `basic_{1,5}shot` and `combo` come from removing responses with a confidence score taken from one of the few-shot examples as described in Section 5.1.

### B.2 Insights into datasets

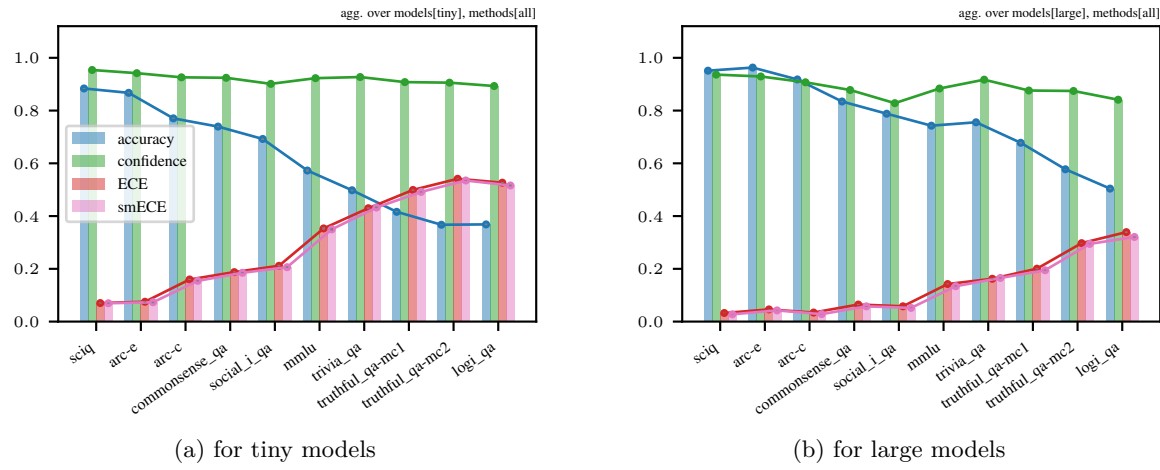

(a) for tiny models

(b) for large models

Figure 7: Calibration per dataset. The metric ECE is defined in Equation (1).

## B.3 Insights into models

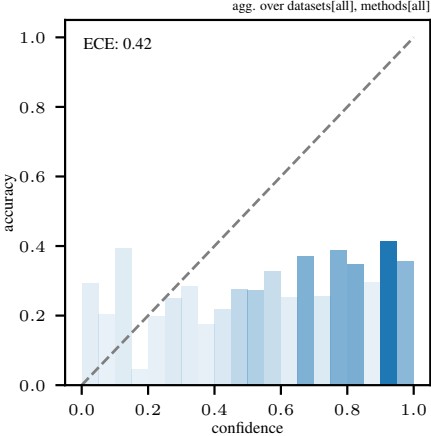

Figure 8: Calibration diagram for `gemma1.1-2b`. The color intensity of each bar is proportional to the bin size on a log scale. Note that the accuracy is close to uniform no matter on which range of confidence scores is conditioned.

## B.4 Insights into prompt methods

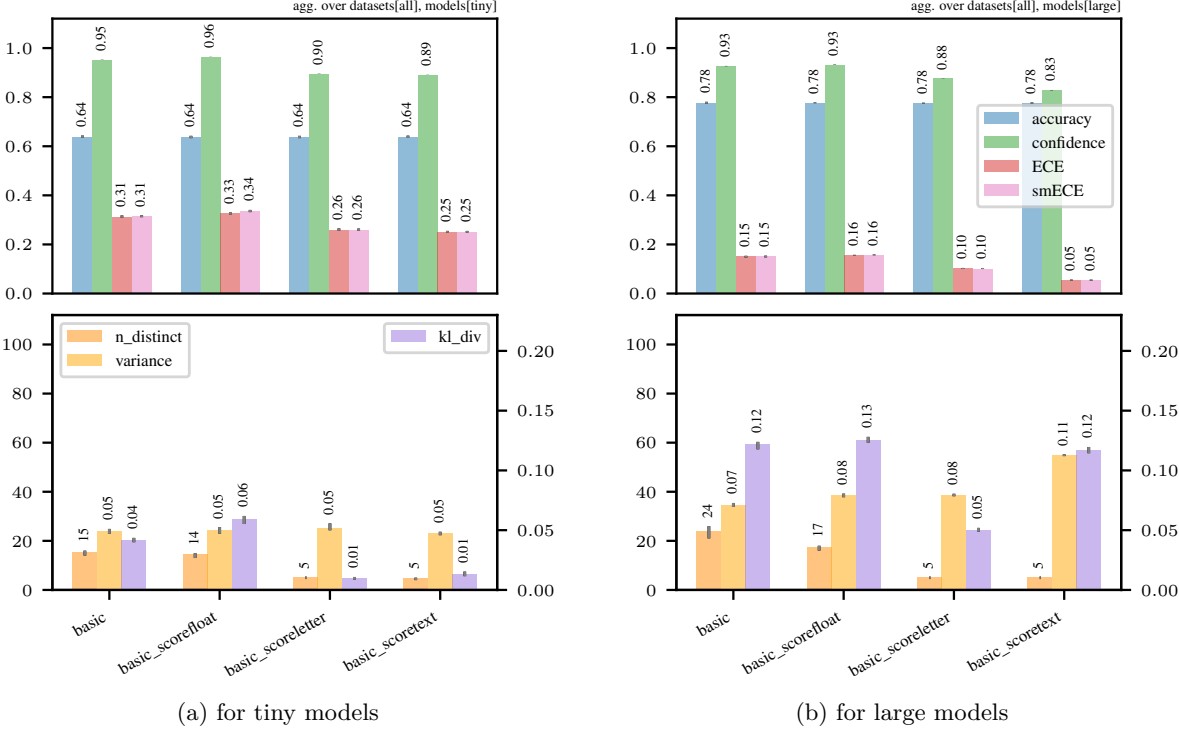

(a) for tiny models                    (b) for large models

Figure 9: Calibration (top), informativeness (bottom) and meaningfulness (bottom) of prompt methods focusing on the aspect "score range". The metrics are defined in Equations (1) to (3).

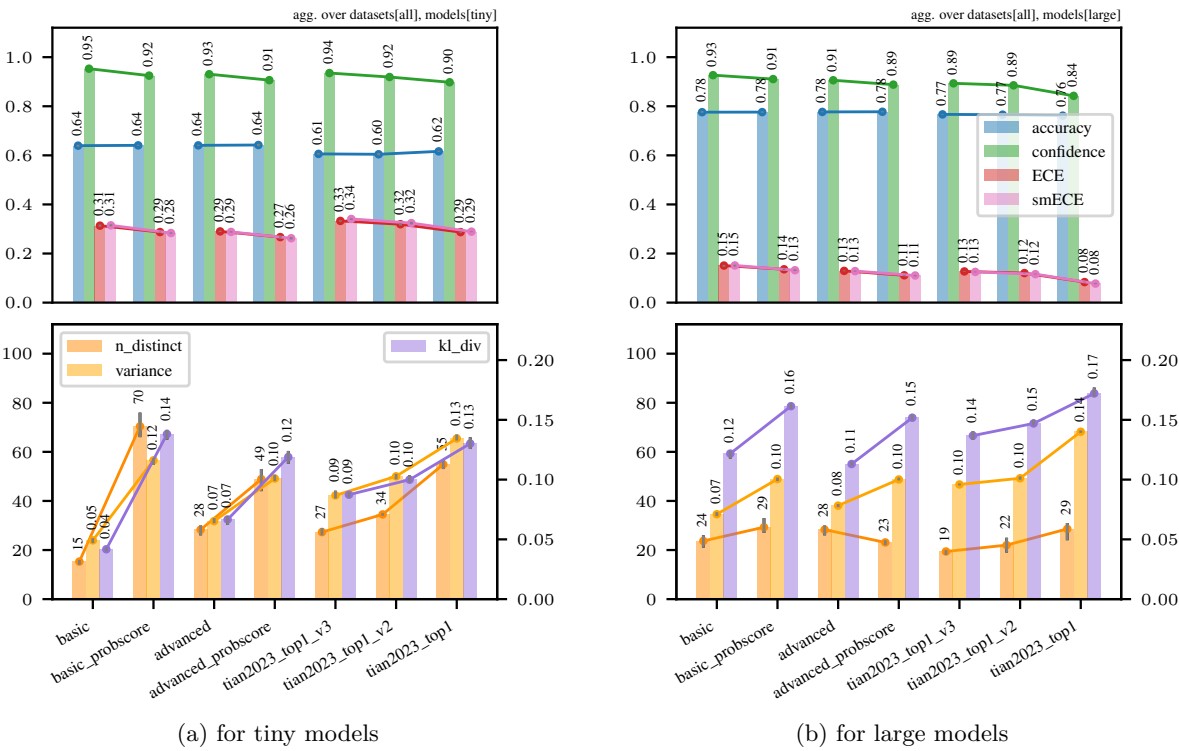

Figure 10: Calibration (top), informativeness (bottom) and meaningfulness (bottom) of prompt methods focusing on the aspect "score formulation". The metrics are defined in Equations (1) to (3).

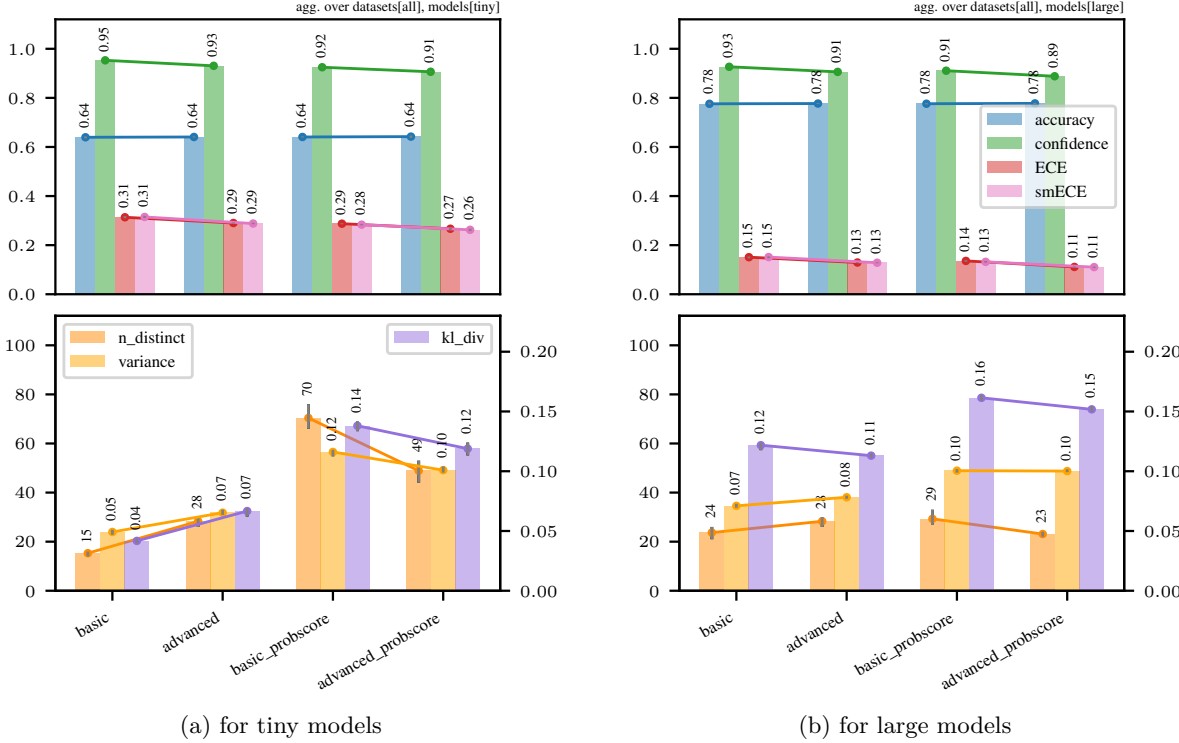

Figure 11: Calibration (top), informativeness (bottom) and meaningfulness (bottom) of prompt methods focusing on the aspect "advanced description". The metrics are defined in Equations (1) to (3).

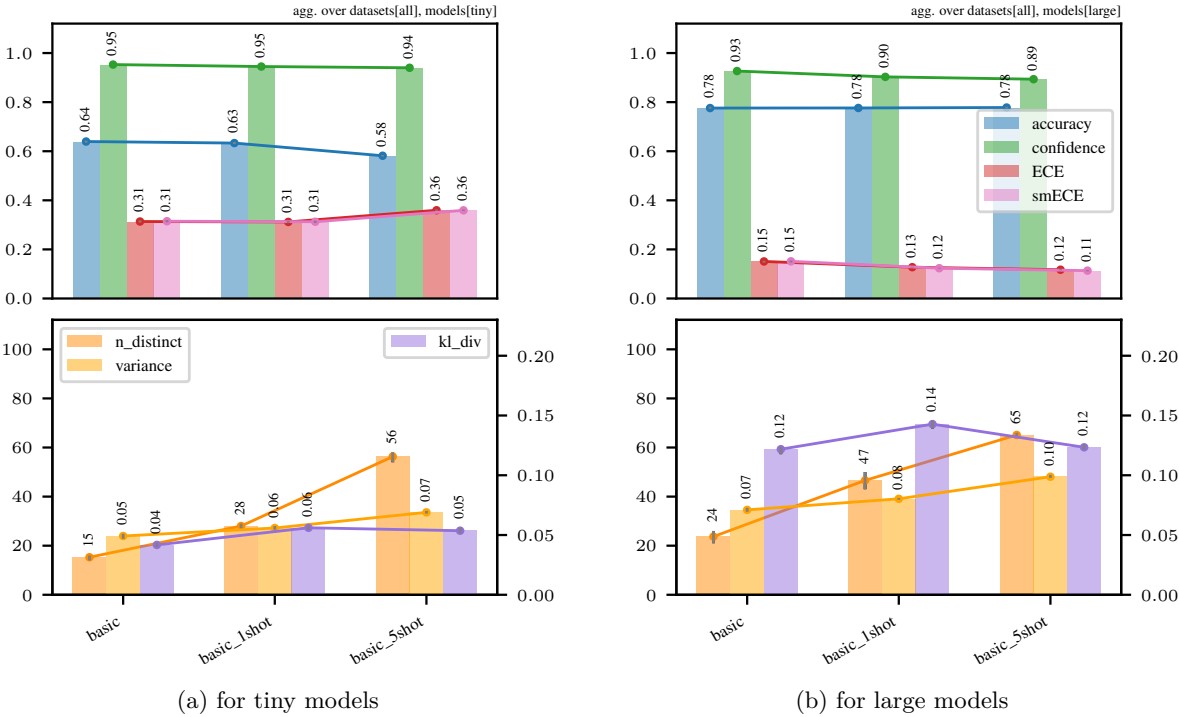

(a) for tiny models  (b) for large models

Figure 12: Calibration (top), informativeness (bottom) and meaningfulness (bottom) of prompt methods focusing on the aspect "few-shot prompting". The metrics are defined in Equations (1) to (3).

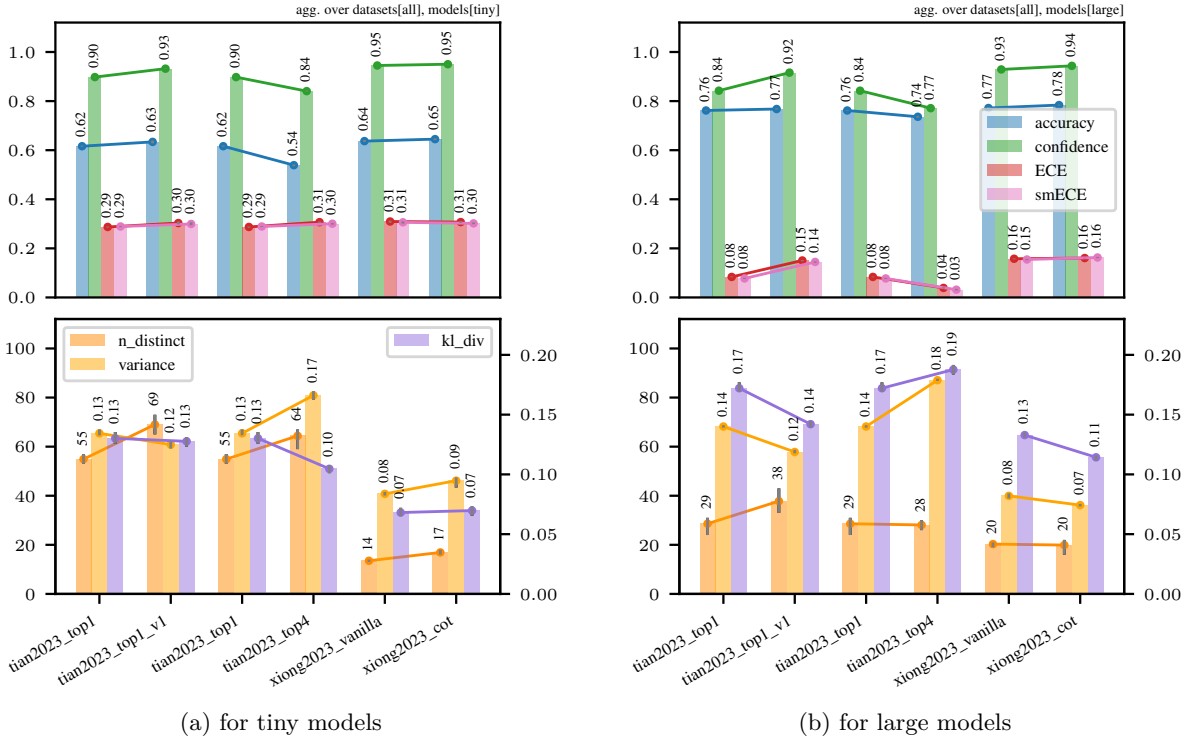

(a) for tiny models  (b) for large models

Figure 13: Calibration (top), informativeness (bottom) and meaningfulness (bottom) of prompt methods focusing on "other aspects". The metrics are defined in Equations (1) to (3).

## B.5 Principled calibration diagrams

While binning-based calibration diagrams such as Figure 5 remain a popular choice for visualizing the ECE, they remain sensitive to the chosen binning strategy. Hence, we additionally evaluate the SmoothECE metric (Blasiok & Nakkiran, 2023) and provide the accompanying smooth calibration diagrams using their `relplot` Python package[4] in Figure 14.

These visualizations offer additional qualitative insights into the calibration performance of tiny and large models. Notably, for GPT-4o, the `combo` method creates a significant separation between the distribution of confidence scores for correct and incorrect responses. This supports our findings in Section 5.4 that the formulation of prompts directly affects the calibration of verbalized confidence scores.

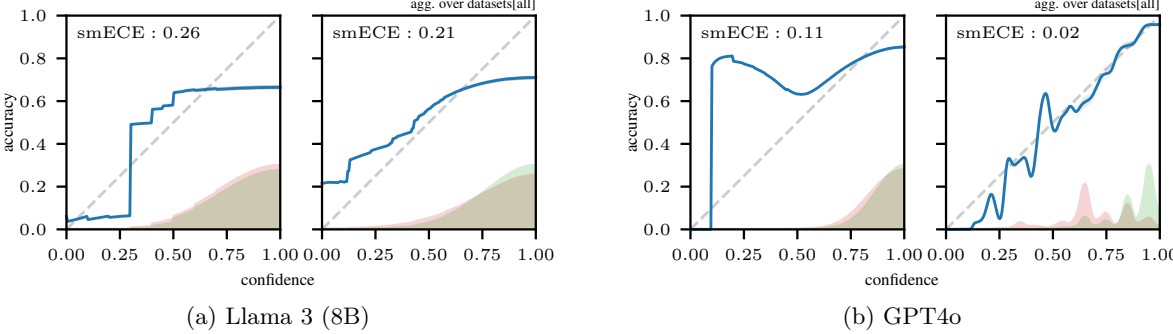

Figure 14: Calibration diagrams for prompt method `basic` (left) and `combo` (right). The blue curve shows the accuracy conditioned on the confidence scores based on the estimated density (Blasiok & Nakkiran, 2023). The green and red areas visualize the underlying density of confidence scores for correct and incorrect responses, respectively, estimated via kernel smoothing. We provide the classical calibration diagrams based on binning in Figure 5.

## B.6 Comparison of calibration metrics and Brier score

In Section 3.2, we introduce the standard notion of calibration and the metric ECE measuring the absolute calibration error (Guo et al., 2017). A common alternative is the Brier score (Glenn, 1950), defined as the expected squared error between the predicted confidence score and the actual outcome:

$$\text{Brier score} = \mathbb{E}\left[(C - \mathbf{1}_{\{Y \text{ is correct}\}})^2\right].\tag{7}$$

As a proper scoring rule, it is minimized only if the predicted confidence scores match the true underlying probabilities. While this is a desirable property, the Brier score does not only measure the calibration, but also takes the informativeness of the confidence scores into account. To show this more formally, the Brier score can be decomposed into three components (Murphy, 1973):

$$\text{Brier score} = \underbrace{\mathbb{E}_C\left[(C - \mathbb{P}(Y \text{ is correct} \mid C))^2\right]}_{\text{calibration}^5} - \underbrace{\text{Var}_C[\mathbb{P}(Y \text{ is correct} \mid C)]}_{\text{resolution}} + \underbrace{\text{Var}\left[\mathbf{1}_{\{Y \text{ is correct}\}}\right]}_{\text{uncertainty}}.\tag{8}$$

The calibration term measures the deviation of the predicted from the true conditional probabilities and resembles a squared analogue to ECE defined in Equation (1). The resolution term quantifies how well the confidence scores can distinguish between correct and incorrect responses, which is commonly referred to as the information content of the confidence scores (Bröcker, 2009). The uncertainty term captures the inherent difficulty in predicting the correctness of a response. Because the Brier score jointly measures calibration and informativeness (i.e., resolution) of the predicted confidence scores, a lower Brier score does not necessarily imply better calibration, as it might result from a higher resolution (Hoessly, 2026, misconception 3).

---

[4]https://github.com/apple/ml-calibration

[5]In the context of weather forecasting, the calibration term is typically referred to as the reliability of the forecast.

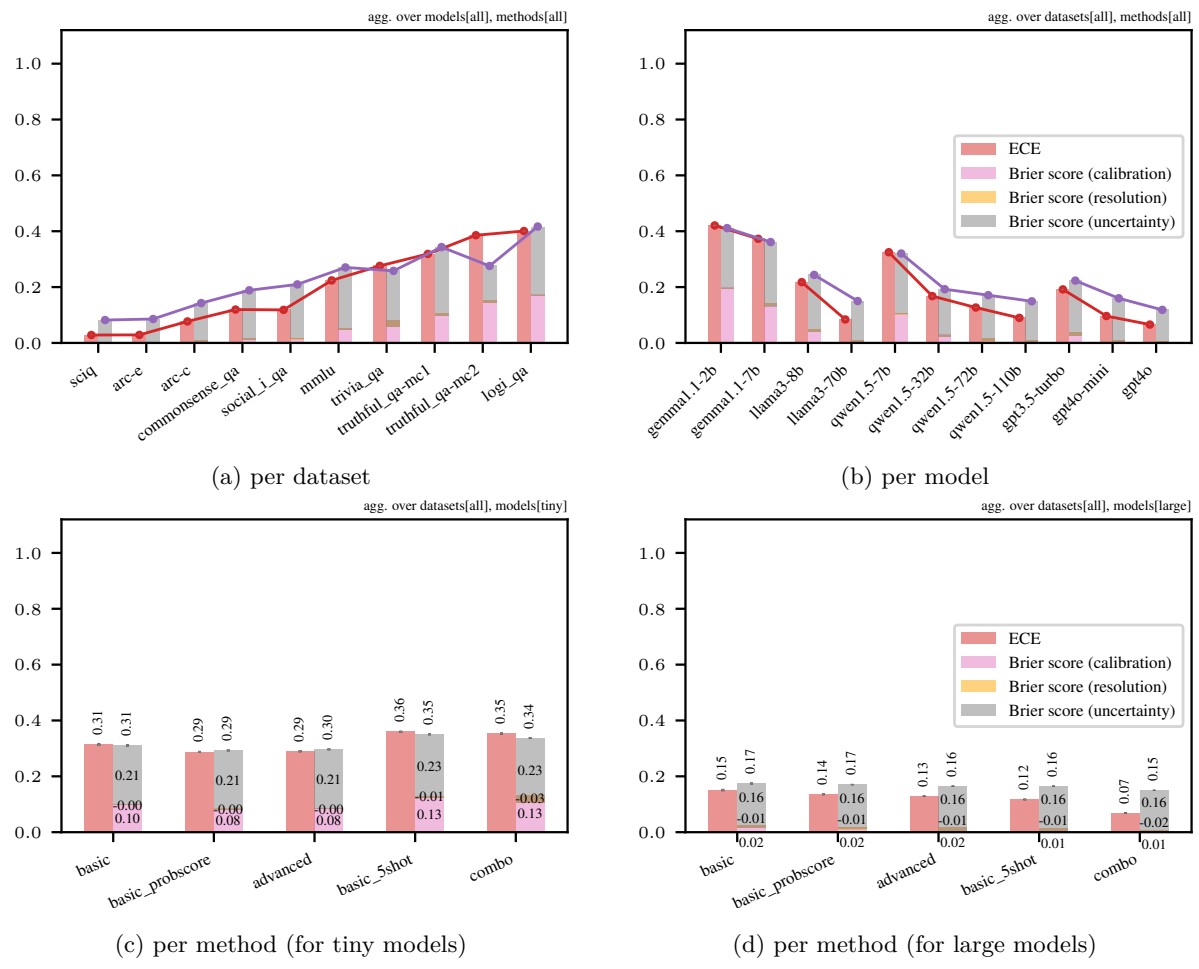

Figure 15: Comparison of ECE with the decomposed Brier score per dataset, model and prompt method. The metric ECE is defined in Equation (1) and the Brier score in Equation (7).

To disentangle these signals, we evaluate calibration and informativeness separately using metrics, which are well-known and easy to interpret, such as ECE in Equation (1) and the variance of confidence scores in Equation (2). In particular, under the assumption of perfect calibration (i.e., $\mathbb{P}(Y \text{ is correct} \mid C = c) = c$) the resolution term becomes equivalent to the variance of confidence scores. This direct relationship motivates the use of the predictive variance as a measure of informativeness to complement ECE.

For completeness, we compare ECE with the decomposed Brier score in Figure 15. As the decomposition in Equation (8) typically relies on binning to estimate the conditional probabilities $\mathbb{P}(Y \text{ is correct} \mid C)$, we adopt the exact binned decomposition from Stephenson et al. (2008). The Brier score exhibits trends consistent with our findings based on ECE in Sections 5.2 to 5.4. The calibration component is directly related to ECE by measuring the squared instead of absolute deviations. The uncertainty component is a direct function of the accuracy in the case of binary correctness,[6] namely $\text{Var}\left[\mathbf{1}_{\{Y \text{ is correct}\}}\right] = \text{acc} \cdot (1 - \text{acc})$. The additional insight is provided by the resolution component, which measures the variance of the true conditional probabilities across different confidence levels. As most models produce heavily concentrated scores as seen in Figure 5, the resolution is generally low. Notably, the marginal increase in resolution observed for the `combo` prompt suggests that this prompt method encourages the model to be more discriminative in its confidence verbalization.

---

[6]For `trivia_qa` and `truthful_qa-mc2`, the correctness label is non-binary due to their answer type as described in Table 1. Hence, their uncertainty components in Figure 15a do not strictly follow this quadratic relationship.

