# OpenReview forum: "On Verbalized Confidence Scores for LLMs"
_TMLR — Rejected by TMLR_

### Review · Reviewer_jDQg · 2025-08-02

**Summary Of Contributions:**

The paper contributes to the field of uncertainty quantification for large language models (LLMs) by focusing on verbalized confidence scores, where LLMs are asked to express their uncertainty as part of their output tokens. It provides an intuitive decomposition of LLM uncertainty into input, model, and output uncertainty, and specifies the reliability of confidence scores through metrics of calibration, informativeness, and meaningfulness. Through extensive experiments across 10 datasets, 11 LLMs, and 17 prompt methods, the paper reveals that the reliability of these scores strongly depends on prompt design, with tiny LLMs benefiting more from simple prompts and large LLMs from complex, combined ones, and identifies a "combo" prompt method that achieves well-calibrated scores for large LLMs.

Strengths
1. The work addresses a critical gap in LLM trustworthiness by focusing on UQ methods that are accessible to end-users and developers, avoiding reliance on internal model states (e.g., token logits) or heavy computational overhead (e.g., multiple response sampling). This aligns with real-world needs for transparent LLM decision-making .
2. The large-scale evaluation (10 datasets, 11 models, 17 prompts) strengthens generalizability. By including diverse model families (open-source vs. closed-source) and datasets (science, commonsense, trivia), the authors avoid overfitting to narrow scenarios .
Clarifying Prior Disagreements: The paper resolves conflicting findings in prior work by attributing discrepancies to prompt methods, providing a clear explanation for why verbalized scores may appear well- or poorly calibrated .
3. Beyond standard calibration (ECE), the authors introduce informativeness (variance of scores) and meaningfulness (KL divergence across datasets) as complementary metrics, offering a more holistic assessment of confidence scores than prior studies.

Weaknesses
1. All datasets are closed-book, closed-ended, and objective. Results may not generalize to open-book (e.g., RAG) or subjective settings, undermining the "prompt-agnostic" claim. How might VCS be adapted for such cases? Could prompt templates include task-specific criteria (e.g., "confidence in creativity")?
2. The work does not systematically compare verbalized scores to established UQ methods (e.g., internal logits, sample consistency, proxy models). Without showing that verbalized scores outperform or complement these methods in key metrics (e.g., calibration, overhead), the case for their superiority remains unconvincing. Meanwhile, discussion of calibration techniques (e.g., temperature scaling, Platt scaling) is missing.
3. The paper identifies that complex prompts benefit large LLMs, but provides no explanation for why (e.g., is it due to better in-context learning, improved understanding of "probability," or reduced overconfidence?).

Minor Comments
1. "let the LLM to self-assess" → "let the LLM self-assess"
2.  "Jiang et al. (2021, Section 4.2) characterizes" → "characterize"

**Audience:**

Yes

**Audience Explanation:**

The findings of this paper might be of interest to at least some individuals in TMLR's audience.

**Claims And Evidence:**

Yes

**Claims Explanation:**

Overall, the key claims are supported by rigorous, large-scale experiments with clear metrics, though some generalizability and explanatory evidence could be strengthened.

**Requested Changes:**

1. Expand dataset scope to include open-book, subjective, and open-ended tasks
2. Systematically compare VCS to established uncertainty quantification (UQ) methods
3. Explain the mechanism behind complex prompts’ efficacy for large LLMs
4. Address minor typos and grammatical inconsistencies

---

> ### Author Response · Authors · 2026-02-18
> **Official Comment by Authors**
>
> Dear reviewer jDQg,
>
> Thank you for your time and effort in reviewing our submission and providing detailed feedback. We would also like to apologize for the long turnaround time due to a delay of several months in the reviewing process, which was not within our control. We are glad that you find our work important for transparent LLM decision-making in the real-world and that you appreciate the comprehensiveness of our experiments.
>
> We would like to address your questions and concerns below:
> - **Regarding dataset scope**: We acknowledge that our evaluation focuses on objective, closed-book datasets, as noted in our limitations section. While expanding to open-book settings (e.g., RAG) is a compelling direction, we were restricted by resource constraints. However, since our study aims to isolate how prompt formulation affects the model's self-assessment of its available knowledge, we expect these influences to persist even when additional context is provided. For subjective or open-ended questions, the challenge lies in the lack of an objective ground truth or a binary definition of "correctness," rendering the evaluation of calibration difficult or impossible. We therefore prioritized tasks with verifiable accuracy to ensure our assessment remains rigorous.
> - **Regarding comparison to other UQ methods**: As noted in our response to all reviewers, our goal is not to claim superior reliability over alternative UQ methods, but rather to investigate the reliability of different verbalized UQ methods themselves. We view the model-agnostic and low-overhead nature of verbalized confidence rather as the motivation for its use. Given that existing work already compares verbalized confidence scores to alternatives, we focused on the granular effects of prompt design.
> - **Regarding calibration techniques**: Classical techniques like temperature or Platt scaling assume that confidence scores are derived from underlying logits. Since verbalized scores are discrete tokens, applying these methods is not straightforward. We would like to refer to Wang et al. (2024), who explore temperature scaling for verbalized confidence scores based on an “invert softmax trick”.
> - **Regarding mechanism behind complex prompts**: While understanding the causal "why" (e.g., improved in-context learning vs. latent calibration) is highly valuable, we believe providing an explanation now would be conjectural. As stated in our response to all reviewers, we focused on establishing the necessary empirical foundation first. We believe that characterizing the phenomenon clearly is a prerequisite for subsequent work on underlying mechanisms.
>
> We have corrected the minor typos you identified. We hope our answers clarify your questions and concerns. Please let us know if there is anything we should elaborate further.
>
> ---
>
> **References**
> - Wang et al. (2024). *Calibrating Verbalized Probabilities for Large Language Models.* arXiv.org. https://doi.org/10.48550/arXiv.2410.06707

---

### Review · Reviewer_BNT5 · 2025-08-22

**Summary Of Contributions:**

The paper focuses on verbalized confidence scores for LLMs, i.e., the LLM verbalizes its own confidence with output tokens. This topic is important as LLMs are widely used in many real applications and it would be very useful to know the confidence of their responses. The paper makes the following contributions:
- provides an framework for uncertainty decomposition for LLMs;
- gives a specification of the reliability of confidence scores for LLMs;
- shows insights into how the dataset difficulty, model capacity, and different prompt methods affect this
reliability;
- releases the code for the experiments.

**Audience:**

Yes

**Audience Explanation:**

This is definitely of interest for TMLR audience.

**Broader Impact Concerns:**

No concerns. The broader impact statement is present and sufficiently address the ethical implications of the paper.

**Claims And Evidence:**

No

**Claims Explanation:**

I think the claims are partly supported by evidence in the paper. The calibration results and the prompt-sensitivity finding are shown clearly in the experimental section. At the same time, the "uncertainty decomposition" is explicitly informal and the experiments restrict to output correctness. so that contribution doesn't seem to be empirically validated.
Their added metrics for "informativeness/meaningfulness" are ad-hoc, and the paper itself notes their effectiveness is debatable. Evidence on prompt methods is mixed: GPT-4o improves sharply under the combo prompt (ECE from 0.10 to 0.02), but looking at Figure 6 it looks like for some models it shows worse calibration than other baselines. Finally, there are no uncertainty intervals or significance tests

**Requested Changes:**

- Can you make the uncertainty decomposition formal?

- Can you use proper scoring rules and report binning sensitivity? E.g., you could add Brier/log loss together with ECE and show robustness to the number of bins

- Can you report CIs (e.g., bootstrap) for ECE/other metrics with multiple seeds?

- Can you better motivate “informativeness/meaningfulness” ?

---

> ### Author Response · Authors · 2026-02-18
> **Official Comment by Authors**
>
> Dear reviewer BNT5,
>
> Thank you for your time and effort in reviewing our submission and providing helpful feedback. We would also like to apologize for the long turnaround time due to a delay of several months in the reviewing process, which was not within our control. We are happy to hear that you find our topic important and useful for LLMs in real applications and that our findings are of interest to the TMLR audience.
>
> We would like to address your questions and concerns below and explain how we have revised our paper in response:
> - **Regarding Figure 6**: We believe there may be a misunderstanding regarding this figure, which visualizes the percentage of valid responses according to our response template, rather than calibration. If you were referring to a different visualization, please let us know so we can provide a specific clarification.
> - **Regarding informal UQ decomposition**: Our goal with this taxonomy is to provide a conceptual framework for the various dimensions of uncertainty in LLM outputs. The intention is to provide a qualitative guide to describe the intuition behind our study's focus, rather than a rigorous mathematical derivation.
> - **Regarding proper scoring rules and confidence intervals**: We appreciate these suggestions. As detailed in our response to all reviewers, we have:
>   - Added **SmoothECE** and principled reliability diagrams (Appendix B.5).
>   - Included a **Brier score** comparison and decomposition (Appendix B.6).
>   - Added **95% confidence intervals** via bootstrapping to all plots.
> - **Regarding motivation for informativeness**: We now provide a formal motivation for our measure of “informativeness”, as noted in our response to all reviewers. Using the Brier score decomposition described in Appendix B.6, we link the variance of predicted scores to the resolution term, under perfect calibration.
> - **Regarding motivation for meaningfulness**: We acknowledge that this is an exploratory concept. We include this as a complementary metric to quantify task sensitivity. Our goal is to measure whether confidence distributions shift appropriately across datasets of varying difficulty. We believe providing such metrics can offer additional perspectives and potentially spark interesting follow-up research.
>
> We hope our answers clarify your questions and concerns. Please let us know if there is anything we should elaborate further.

---

### Review · Reviewer_K5iQ · 2026-02-06

**Summary Of Contributions:**

This paper evaluates the accuracy of verbalized confidence scores on a series of LLMs.
The authors prompt language models in a few different ways to extract a *verbalized confidence*.
Results are reported in graphs in the main paper and appendix, with some discussion of the empirical observations.

**Audience:**

No

**Audience Explanation:**

In the papers current state, I think it would not be of interest to the TMLR community. Fig. 3 is the most interesting result, though the findings are not particularly surprising.

1. It evaluates a series of obsolete language models.
2. The metrics of *Informativeness* and *Meaningfulness* (which are very poorly named, given they are already well defined mathematical quantities) are confusing. There could be some value in evaluating diversity, but it is not done well here.
3. The authors use ECE, which is hard to compare against, due to its parametric nature. *
4. A general lack of comprehensiveness in both the experiments and the analysis.

*please read [1] to learn about better ways to measure calibration error.

[1] Arrieta-Ibarra I, Gujral P, Tannen J, Tygert M, Xu C. Metrics of calibration for probabilistic predictions. Journal of Machine Learning Research. 2022;23(351):1-54.

**Broader Impact Concerns:**

I think this paper doesn't have any major safety implications.

**Claims And Evidence:**

Yes

**Claims Explanation:**

The authors primarily comment on the empirical observations of the experiments that they run, and don't over-generalize.

**Requested Changes:**

I think some version of this paper could pass the bar of "audience interest" if the authors did the following:

1. Did a comprehensive analysis of many models. Covering proprietary models, open source models, vision-language models, reasoning vs. no reasoning. -- identify any systematic trends in calibration of verbalized confidence.
2. Include more extensive and harder datasets.
3. Investigate different prompting strategies, and dig much deeper into this. A very thorough analysis of strategies could make this paper significantly more interesting.
4. Compare with proxy models and sampling consistency approaches (runtime, other metrics).
5. Use a better calibration analysis tool.

Overall, I think the changes would essentially require a complete re-write of this paper, and require significantly more effort.

---

> ### Author Response · Authors · 2026-02-19
> **Official Comment by Authors**
>
> Dear reviewer K5iQ,
>
> Thank you for your time and effort in reviewing our submission and providing critical and honest feedback. We appreciate the high bar set for TMLR and have used your comments to strengthen our work.
>
> We would like to address your questions and concerns below and explain how we have revised our paper in response:
> - **Regarding extension to more recent models**: As noted in our response to all reviewers, there has been a significant delay in the reviewing process since our submission in July 2025. At that time, our evaluation was reasonably up-to-date, including models through July 2024 (e.g., GPT-4o-mini). While the model landscape has rapidly evolved since then, we believe our results remain highly relevant for three reasons:
>   - **Historical baseline**: Our study provides a rigorous "historic checkpoint" of verbalized confidence behavior across 11 different open-source and proprietary models, offering a baseline for future comparative work.
>   - **Resolving conflicts in literature**: One of our core contributions is demonstrating that conflicting findings in recent literature (e.g., regarding whether LLMs are inherently well-calibrated) are often artifacts of sub-optimal prompting. This insight, that the prompt method is a critical factor for the reliability, remains an important takeaway for the community.
>   - **Empirical contribution**: We provide a comprehensive empirical foundation that was previously lacking in the field. While the importance of prompting is often intuitively assumed, our work is the first to systematically quantify how diverse prompt methods directly impact the reliability of verbalized uncertainty.
> - **Regarding extension to harder datasets**: Our evaluation includes a diverse range of task difficulties, as seen in Figure 3. We focused on objective, closed-book tasks to ensure a verifiable ground truth. For subjective or open-ended tasks, calibration becomes methodologically difficult to measure due to the lack or difficulty of evaluating the ground truth. We chose to prioritize a rigorous assessment based on verifiable tasks rather than introducing the noise associated with subjective evaluation.
> - **Regarding extension to more prompt methods**: Our selection of 17 prompt methods was designed to systematically isolate key variables including score range, linguistic formulation, few-shot effects, and descriptive complexity. Rather than an ad-hoc collection, this represents a comprehensive analysis aimed at identifying which specific prompt components most significantly impact reliability. While the space of possible prompts is vast, our study establishes the necessary empirical bounds to guide future investigations into more niche or specialized prompt methods.
> - **Regarding lack of comprehensiveness**: While no study can be exhaustive, our evaluation encompasses 10 datasets, 11 models, and 17 prompt methods, totaling nearly 2 million generated responses. To our knowledge, this represents one of the largest systematic studies of verbalized confidence scores to date. We believe this scale is necessary to resolve the conflicting findings in current literature (Xiong et al., 2023; Tian et al., 2023), which we demonstrate are heavily influenced by the specific prompting strategies used.
> - **Regarding confusing notion of informativeness/meaningfulness**: We appreciate the critique regarding terminology. As noted in our response to all reviewers, we now provide a formal motivation for our measure of “informativeness”. Using the Brier score decomposition described in Appendix B.6, we link the variance of predicted scores to the resolution term, under perfect calibration. Regarding meaningfulness, we use it as an exploratory metric for quantifying task sensitivity, measuring whether a model’s confidence distribution shifts appropriately under varying task difficulty. We believe providing such metrics can offer additional perspectives and potentially spark interesting follow-up research.
> - **Regarding flaws of ECE**: We agree with your concerns regarding ECE. In the revision, we have included SmoothECE (Blasiok & Nakkiran, 2023) in our main plots and principled reliability diagrams in Appendix B.5, alongside a comparison with the Brier score in Appendix B.6. We find that our core findings remain consistent across these more robust metrics.
> - **Regarding comparison with alternative UQ methods**: As further elaborated in our response to all reviewers, our study focuses on the reliability of verbalized confidence itself. Given that existing work already compares verbalized confidence scores to alternatives, we focused on the granular effects of prompt design, which has been often overlooked by broader comparisons.
>
> We hope we were able to address your concerns and demonstrate the relevance and rigor of our comprehensive evaluation. Please let us know if there is anything we should elaborate further.

---

### Author Response · Authors · 2026-02-18
**Official Response to all Reviewers (part 2)**

We appreciate the reviewers' suggestions to extend this study. While these are valuable directions, some fall outside our primary scope. Our core objective is to provide a systematic empirical analysis of how prompt design affects the reliability of verbalized confidence. This focus is necessitated by conflicting findings in current literature (Xiong et al., 2023; Tian et al., 2023), where we demonstrate that prior disagreements often stem from sub-optimal prompting rather than inherent model limitations. With this focus in mind, we address each of the specific requests below:
- **Explain mechanism behind complex prompts**: We agree that understanding the exact mechanisms behind verbalized confidence scores is highly valuable. While our study provides empirical evidence for prompt dependence, identifying the causal "why" (e.g., improved in-context learning vs. latent calibration) would currently be conjectural. To avoid over-claiming or over-generalizing our results, we have focused on providing the empirical foundation necessary for such future investigations. We believe that characterizing the phenomenon clearly is a prerequisite for subsequent work on the underlying mechanisms.
- **Extend to harder datasets**: We acknowledge that our evaluation focuses on objective, closed-book datasets, as noted in our limitations section. While open-book settings are a compelling direction, we were restricted by resource constraints. For subjective or open-ended questions, the challenge is primarily methodological. Without an objective ground truth or a binary definition of "correctness" (e.g., define “correctness” for a creative poem) evaluating calibration becomes difficult or impossible. While LLM-as-a-judge could serve as a proxy, its (computational) cost and inherent imperfections led us to prioritize verifiable accuracy to ensure a rigorous assessment. Nevertheless, we believe that our insights on closed-form tasks already provide a valuable proof-of-concept for future extensions.
- **Extend to more recent models**: One reviewer asked for a more comprehensive analysis across more recent models. We can understand, given a substantial delay in the reviewing process since our initial submission to TMLR in July 2025 and the rapid explosion of newer and better models every month, that our evaluations seem to be out-dated. Still, we believe that our results and publicly available evaluation framework are of value to the research community, as they not only provide a historic checkpoint of the reliability of verbalized confidence scores for past models, but might also spark follow-up work continuing to assess the reliability of verbalized confidence scores on newer models.
- **Compare with non-verbalized UQ methods**: We agree that a direct comparison with alternative UQ methods would be of interest. However, our study focuses on assessing the reliability of verbalized confidence itself, instead of proving its superiority over alternative methods. We view its prompt-agnostic, model-agnostic and efficient nature rather as the rationale for its relevance, but not as a claim for its overall superiority. Since cross-comparisons with alternative methods are already well-documented (e.g., Lin et al., 2022; Kadavath et al., 2022; Xiong et al., 2023), we intentionally focused on how prompt design specifically affects the reliability, which broader cross-method comparisons often overlook.

We hope that we were able to address all concerns of the reviewers and convinced them of the significance of our empirical findings based on improvements we made to the revised version. While certain requested extensions remain outside our current scope, we believe that this study provides a necessary foundation for the community to build upon.

---

**References**
- Blasiok & Nakkiran. (2023). *Smooth ECE: Principled Reliability Diagrams via Kernel Smoothing.* ICLR 2024. https://openreview.net/forum?id=XwiA1nDahv
- Bröcker. (2009). *Reliability, sufficiency, and the decomposition of proper scores.* Quarterly Journal of the Royal Meteorological Society. https://doi.org/10.1002/qj.456
- Kadavath et al. (2022). *Language Models (Mostly) Know What They Know.* arXiv.org. https://doi.org/10.48550/arXiv.2207.05221
- Lin et al. (2022). *Teaching Models to Express Their Uncertainty in Words.* Transactions on Machine Learning Research. https://openreview.net/forum?id=8s8K2UZGTZ
- Murphy. (1973). *A New Vector Partition of the Probability Score. Journal of Applied Meteorology and Climatology.* https://doi.org/10.1175/1520-0450(1973)012%253C0595:ANVPOT%253E2.0.CO;2
- Tian et al. (2023). *Just Ask for Calibration: Strategies for Eliciting Calibrated Confidence Scores from Language Models Fine-Tuned with Human Feedback.* EMNLP 2023. https://doi.org/10.18653/v1/2023.emnlp-main.330
- Xiong et al. (2023). *Can LLMs Express Their Uncertainty? An Empirical Evaluation of Confidence Elicitation in LLMs.* ICLR 2024. https://openreview.net/forum?id=gjeQKFxFpZ

---

### Author Response · Authors · 2026-02-18
**Official Response to all Reviewers (part 1)**

Dear reviewers,

We would like to thank all of you again for your constructive feedback, which helped us to significantly improve our paper. We have uploaded a revised version where we highlighted the differences from the initial submission in **purple**.

Here, we provide a summary of your main concerns and our changes we made to the revised version.
- **Provide confidence intervals**: As suggested by reviewer BNT5, we re-sampled our evaluation dataset multiple times using bootstrapping and updated all plots with 95% confidence intervals. Please note that we evaluate the metrics on the predictions aggregated over multiple dimensions (e.g., 11 models × 17 methods × 1,000 samples = 187k predictions per dataset in Figure 3a). Due to this large number of evaluation samples, the confidence intervals are typically rather narrow (e.g., ±0.001) and barely visible. We believe that this is a strength of our evaluation setup, demonstrating the statistical significance of our results.
- **Use a better measure for calibration**: We are thankful for the reviewers pointing out the sensitivity of the metric ECE to the binning strategy. Consequently, we additionally report the metric SmoothECE (Blasiok & Nakkiran, 2023), a non-parametric extension of ECE based on kernel smoothing with an automatically chosen kernel bandwidth, in all plots. This metric has been shown to provide better theoretical guarantees than the classical ECE. We observe that SmoothECE follows consistent trends as ECE and that our initial findings based on ECE are not invalidated by its underlying flaws, likely due to our large sample size. In addition, we provide more principled calibration diagrams using their Python package [relplot](https://github.com/apple/ml-calibration) in Appendix B.5, which further strengthen our qualitative insights in the improved calibration behavior when using the prompt method combo for GPT4o.
- **Use proper scoring rules**: To further strengthen our insights, we added a comparison between ECE and the Brier score in Appendix B.6. We highlighted our rationale of not choosing the Brier score as our main metric for calibration, as it jointly measures calibration and resolution and, hence, a lower Brier score does not necessarily imply better calibration. However, using the well-known decomposition of the Brier score (Murphy, 1973), we are able to disentangle the calibration from the resolution, with the latter being colloquially referred to as the “informativeness” of the confidence scores (Bröcker, 2009). In our additional plots, we observe consistent behavior between ECE and the Brier score and provide additional insights based on the resolution component. We would like to thank the reviewers for pointing us in this direction for a more holistic evaluation of the reliability of confidence scores.
- **Improve motivation of informativeness**: Based on the decomposition of the Brier score into a calibration, resolution and uncertainty term, we can formally motivate our choice of the variance of confidence scores as a measure of informativeness in Appendix B.6, complementing the calibration metrics. In short, the resolution term is commonly described as capturing the “informativeness” of the predicted confidence scores (Bröcker, 2009) and, under the assumption of perfect calibration, is equivalent to the variance of predicted confidence scores. Both underlie the same intention to quantify how expressive the predicted confidence scores are (e.g. always predicting 0.5 vs. predicting 0.9 on the correct and 0.1 on the incorrect responses).

---

### Decision · Action_Editor_B5ZH · 2026-04-27

**Recommendation:** Reject

**Additional Comments:**

Two out of the three reviewers argue strongly for rejection even after the rebuttal while the remaining reviewer is mildly positive. Based on my reading of the reviews and the rebuttal, the paper is not suitable for publication at TMLR.

**Audience:**

Yes

**Audience Explanation:**

Two reviewers think at least some individuals iN TMLR's audience be interested in knowing the finding of this paper but one reviewer thinks otherwise. Based on the paper content and the reviews, I think there are sufficient reasons for a yes.

**Claims And Evidence:**

No

**Claims Explanation:**

Two reviewers think the claims in the submissions are not supported by accurate, convincing and clear evidence but one reviewer thinks otherwise. Based on the paper content and the reviews, I think there are sufficient reasons for a no.